# OpenReview forum: "KTAD: Knowledge Trust-Aware Adaptive Dropout for Enhanced CNN Regularization"
_ICLR.cc/2026/Conference — Submitted to ICLR 2026_

### Official Review · Reviewer_5iGu · 2025-10-28

**Soundness:** 2
**Presentation:** 1
**Contribution:** 2
**Rating:** 2
**Confidence:** 3

**Summary:**

This paper considers improving the conventional dropout technique by making the dropout rate per convolutional layer in a CNN model be sample and time dependent. To compute the adaptive dropout rate for each convolutional layer, a sequence of processing steps are performed, which include processing via two-layer MLP (corresponding to (2)), softmax operation (corresponding to (3)) for computing the attention per location in a 2d space,  LayerNormaliztion, one-layer MLP (corresponding  to (5)), and incorporating channel-wise statistics and adaptive scheduling strategies. A number of experimental results are presented for CNN models showing the advantages of the new dropout method.

**Strengths:**

(1) I think one main strength is that channel-wise statistics are computed and incorporated when computing the adaptive dropout rate.
(2) The second strength is that the authors manage to make the new dropout method work via a long sequence of processing steps.

**Weaknesses:**

(1) I think the new dropout method is too complicated, it is not clear which component makes a major contribution without ablation study. Given that the new method consists of many steps, I think the ablation study is also difficult to conduct.

(2) There are inconsistencies between the introduction and the main paper. In introduction, the authors claim that "KTAD leverages these scores to apply higher dropout rates to unreliable channels while preserving informative ones". In Section 3, only a single dropout rate is computed per sample per CNN layer, which is given by (11).

(3) The notations are not consistent in the paper. It is not clear to me which is the final expression for t_b. Is it (5) or (9)?


(4) The validation accuracy for CIFAR100 is far from SOTA. See the reference below.

(5) The performance improvement in the experimental section is not significant.

* open-source for training resent18 over CIFAR100: https://github.com/weiaicunzai/pytorch-cifar100

**Questions:**

see my comments above.

---

### Official Review · Reviewer_hCQb · 2025-10-30

**Soundness:** 2
**Presentation:** 1
**Contribution:** 2
**Rating:** 2
**Confidence:** 4

**Summary:**

This article introduces a drop-in replacement that computes trust-aware adaptive dropout to preserve information features while more aggressively reglazing weaker ones, achieving higher classification accuracy for image classification problems, compared to baseline methods such as DropBlock. Theoretical results show that the adaptive dropout scheme leads to lower gradient variance, faster convergence and tighter generalization bounded. Moreover, model robustness and computational efficiency are also improved.

**Strengths:**

-	The idea of developing channel dependent dropout rate based on channel attention and activation statistics is interesting.

**Weaknesses:**

-  The methodology and theoretical parts are poorly written.
- The numerical evaluation is not conclusive enough.

**Questions:**

- The step 3 in the section 3.2 computes a per-sample trust score t over a mini-batch sample of size B. It is not clear how this score is related to channels.
- Section 3.6 is hard to follow since the basic definitions of L and err_gen and err_train are not introduced. The statement of Corollary is unclear, and there is no proof about it.
- It is hard to understand the results in Table 2. Why KTAD accuracy on Imagenet-32 is higher than the one in Table 1? The baseline performance of DropCluster on ImageNet is not missing.
- Why the attention maps have a different size compared to the input image in Fig 3?

---

### Official Review · Reviewer_vFGR · 2025-11-01

**Soundness:** 2
**Presentation:** 2
**Contribution:** 2
**Rating:** 2
**Confidence:** 4

**Summary:**

This paper proposes KTAD (Knowledge Trust-Aware Adaptive Dropout), a novel adaptive dropout method for CNNs. Unlike conventional static dropout methods, KTAD dynamically calculates a 'trust score' to evaluate the reliability of each data sample.
This score is leveraged to modulate the dropout intensity on a per-sample basis—applying less regularization to high-trust samples and more to low-trust ones—which is combined with a global scheduling strategy. The authors provide theoretical proofs that KTAD achieves lower gradient variance and faster convergence, and experimentally demonstrate its superior performance over existing methodologies on several image benchmarks.

**Strengths:**

- The proposed method demonstrates superior performance, consistently outperforming existing dropout variants across multiple datasets.
- The paper also provides theoretical evidence (e.g., Theorem 1, 2) to support its claims.
- The core claims are intuitive, and the paper is easy to read.

**Weaknesses:**

- Overall weakness: The theoretical proofs and experimental analyses are not clear. The authors tend to just list performance results from their experiments, rather than providing a deep analysis of why the proposed methodology achieves superior performance. Furthermore, the paper would be more comprehensive if it included details on whether the method is applicable to fully connected layers and its corresponding performance (although Table 9 seems to suggest this is possible).

Section 3
- Section 3.2, Line 123, equation (2):There is no mention of why the hidden layer size of the weight matrix was set to 64. Furthermore, the explanation regarding the initialization of $W_1, W_2, w_f$ is missing, leaving it unclear whether they were initialized in the same manner as other standard linear weights.
- Line 129: It would be more intuitive to explicitly state that the shape of $\alpha^\top$ is $B\times 1 \times (HW)$ and that the shape of $F_{att}^t$ becomes $B\times C$ after being squeezed from $B\times 1 \times C$.
- Line 132: The paper does not explicitly state that the $\sigma$ function is the sigmoid function.

Section 3.3
- Line 150: It is questionable whether $u_c^{(t)}$ can be called a "bounded stability score." If $\mu$ is negative and $\sigma$ is close to 0, $u$ will be close to 0, but $x$ can be considered stable. If $x$ is a value that has passed through a ReLU, $u$ would be calculated as $u \ge 0.5$, and an explanation is needed as to whether this is the intended calculation.
- Line 155: Reading only the main text, one might wonder why $\alpha=0.9$ was chosen. Since the relevant experiment is provided in the Appendix, it would be beneficial to explain the choice of momentum 0.9 with a reference to it.
- The term "temperature" appears frequently in KTAD (equations (5), (9), and (11)). There is no explanation as to whether these temperatures all share the same value or what values they were set to. The temperature mentioned in the experiments seems to be the one from equation (11), but clarification on the other temperatures is needed.

Section 3.6
- The proofs for the Theorems are only provided as proof sketches, and the full, practical proofs cannot be found. The content in Appendix A.6 seems to be unrelated to these proofs.
- The assumption in Theorem 1, "when trust scores are well-calibrated," seems potentially too strong.
- While Theorem 2 and Appendix 2.7 briefly show that the training convergence speed is faster, the claim would be much stronger if an Accuracy Learning Curve graph comparing KTAD with other methods were provided.

Section 4
- As mentioned in the Limitations, the results on ResNet-18 alone may not be sufficient to fully substantiate the paper's claims. Adding experimental results on Vision Transformer or VGG models would significantly enhance the paper's persuasiveness.
- It is a limitation that the experiments for other baselines were set up only as described in their respective papers. For instance, DropCluster and KTAD use the Adam optimizer, while Very-Large Dropout, Stochastic Depth, and Adaptive Dropout use SGD with momentum. If hyperparameters such as weight decay or dropout rates were varied, the experimental results for the baselines could potentially change.

Section 5
- For each dataset, only the strongest baseline is presented in the table. The results against other baselines are not found in the Appendix, making it difficult to verify the full comparison.
- An explanation of GFLOP and Win Rate is absent. Does the win rate mean the models were trained with 200 different seeds and compared against each other?

Section 5.1
- Can a higher win rate truly be equated to model robustness? I generally understand robustness to mean maintaining performance when noise is added to the input, but this seems to be used in a different context here. An explanation of this is necessary.

Section 5.3
- The method for calculating computational efficiency and an explanation of the results are missing.

Section 5.4
- As mentioned above, there is a suspicion that Figure 3 was generated using an LLM. Furthermore, there are discrepancies with the description in the main text. For example, in the Airplane case, both the Baseline and KTAD have attention on the sky, and in the Ship case, the attention heatmap is generated on the ship's wake. These are peculiar results that warrant further explanation, and the lack thereof is a significant drawback.

Appendix
- Line 676, A.2.2: Typo, "Exponential decay" → "Cosine annealing".
- A.2.3: In Table 7, for ImageNet-34, the bolding should be on temperature 0.5.
- A.2.4: A description of the "Base CNN" is missing. It is important to know if the Base CNN uses standard dropout, but this information is absent. KTAD requires an additional $66C+64$ learnable parameters per layer (though this count might not be accurate) and also requires softmax and layer normalization, yet the training time is reported to be almost unchanged compared to the Base CNN. This seems questionable. Additional clarification on this point would be necessary.
- A.2.5: A description of the CNN architecture used in this experiment is also missing. Since the late conv layers are at the 5th-6th position, it is presumed not to be ResNet-18. It is also unclear if the trust score for the FC layers is calculated as in Section 3.2.
- A.6: The content here seems to argue that a single mask is better for KTAD than a per-channel approach. However, the proof for this is very weak, and it is questionable why this content exists independently in the appendix without being referenced in the main body of the paper.
- Lines 1060, 1070: Typo, "Lemmas 20-24" should be "Lemmas 2-3".

**Questions:**

Already mentioned in Weaknesses part.

---

### Official Review · Reviewer_Yzki · 2025-11-05

**Soundness:** 2
**Presentation:** 2
**Contribution:** 1
**Rating:** 2
**Confidence:** 4

**Summary:**

The authors present a method that stocastically drops input examples according to how trustworthy they are deemed to be.

Inputs are dropped based on trust scores, which are determined by combining the predictions (eq. 9) of a trained network (eq. 1-5), and averaged data statistics, as described in (eq. 6-8). Note that while per-channel responses and statistics are intermediate outputs of their network and average statistics, repectively, the final trust scores and corresponding dropout rates are set on a per-example basis. In addition to trust-score adjustments, the dropout rate is moderated by an adaptive schedule (eq. 10),
 and a temperature scale (eq. 11).

Experimental results using CNNs on SVHN, CIFAR-100-C, CIFAR-100, and ImageNet-32 suggest that their approach slightly outperforms well-known dropout baselines (Table 1, 2).

**Strengths:**

- The approach seems sound.
- The reported results indicate small gains over popular dropout baselines.

**Weaknesses:**

- The abstract and other significant portions of the paper need significant revision, as they are misleading. While there are intermediate elements of their network-based and statistics-based trust scores that are computed on a per-channel basis, ultimately dropout is applied at example level. There is commentary explaining this later in the paper, but this needs to be more clearly articulated throughout.
- Since "channel-awareness" is a major theme that is discussed often throughout the paper, explanations, ablations and analysis to substantiate these claims should be included, as it is not clear that this is the case.
- As an data filtering/reweighting method, this approach should be compared to other sota data-filtering/reweighting methods, not dropout approaches. This needs to be resolved before the paper can be considered for publication.
- The trust network must be trained, but no information regarding how this was done was included. What targets were utilized? How were they obtained?
- The analysis in section 3.6 suggesting that appropriate input-adaptive dropout lowers variance is compelling, but in this case the example dropout probabilities are no longer independent, which is not taken into account.

**Questions:**

See previous section.

---

### Meta-Review · Area_Chair_GJV5 · 2026-01-06

**Summary:**

Four clear rejections.

**Reviewer Concerns:**

Four clear rejections. This is no rebuttal.

**Reviewer Scores:**

Four clear rejections.

---

### Decision · Program_Chairs · 2026-01-26

Reject